# Quality Attributes and Metabolic Profiles of Uvaia (*Eugenia pyriformis*), a Native Brazilian Atlantic Forest Fruit

**DOI:** 10.3390/foods12091881

**Published:** 2023-05-03

**Authors:** Poliana Cristina Spricigo, Luísa Souza Almeida, Gabriel Henrique Ribeiro, Banny Silva Barbosa Correia, Isabela Barroso Taver, Angelo Pedro Jacomino, Luiz Alberto Colnago

**Affiliations:** 1Luiz de Queiroz College of Agriculture, University of São Paulo, 11 Pádua Dias Ave., Piracicaba 13418-900, São Paulo, Brazil; polianaspricigo@usp.br (P.C.S.);; 2School of Agricultural and Veterinarian Sciences, São Paulo State University, Jaboticabal 14884-900, São Paulo, Brazil; 3Institute of Chemistry of São Carlos, University of São Paulo, 400 Trabalhador São Carlense Ave., São Carlos 13566-590, São Paulo, Brazil; 4Embrapa Instrumentation, 1452 XV de Novembro Street, São Carlos 13560-970, São Paulo, Brazilluiz.colnago@embrapa.br (L.A.C.); 5Department of Food Science, Aarhus University, 48 Agro Food Park, 8200 Aarhus, Jutland, Denmark

**Keywords:** *Eugenia pyriformis*, nuclear magnetic resonance, postharvest

## Abstract

The uvaia is a native Brazilian Atlantic Forest Myrtaceae fruit with a soft pulp, ranging from yellow to orange, with a sweet acidic flavor and sweet fruity aroma. Uvaias present consumption potential, but their physicochemical characteristics are still understudied. In this context, we describe herein the metabolites of uvaia that have been determined by nuclear magnetic resonance spectroscopy. We screened 41 accessions and selected 10 accessions based on their diversity of physicochemical attributes, i.e., their fresh mass, height, diameter, yield, seed mass, total soluble solids, and titratable acidity. Twenty-six metabolites were identified, including sugars, acids, and amino acids. The results of this study comprise the most complete report on sugars and acids in uvaias. The relevant metabolites in terms of abundance were the reducing sugars glucose and fructose, as well as malic and citric acids. Furthermore, this study represents the first description of the uvaia amino acid profile and an outline of its metabolic pathways. Uvaia quality attributes differ among accessions, demonstrating high variability, diversity, and several possibilities in different economic areas. Our findings may help in future breeding programs in the selection of plant material for industries such as food and pharmaceuticals.

## 1. Introduction

Native Brazilian Atlantic Forest fruits have received significant attention regarding their quality, sensory attributes, and chemical composition [1,2,3,4], and are used in both the food sector and as a raw material source for ingredients that are employed in different industries. The uvaia (*Eugenia pyriformis*) is a native Atlantic Forest fruit belonging to the Myrtaceae family, a widespread family in the tropics that is distinguished by its leathery leaves that are filled with oil glands [5]. Uvaia stands out for its wide variety of shapes, from round and flattened to elongated and pyriform, and colors, ranging from light yellow to darker orange shades. Its color directly relates to the presence of the carotenoids b-carotene, b-cryptoxanthin, a-carotene, lutein, and zeaxanthin [6]. α-carotene, β-carotene, and β-cryptoxanthin are precursors of vitamin A, which is an essential vitamin for humans.

The most remarkable uvaia features are its fruity note aroma and phenolic and terpene contents [7]. Its volatile profile has 59 identified compounds, with a predominance of monoterpenes and sesquiterpenes [4,7]. Gallic acid has been reported as the main phenolic component in this fruit [8]. Terpenes have received attention for their health benefits, such as exerting anti-inflammatory effects by inhibiting the pro-inflammatory pathway, anti-tumorigenic effects on various systems both in vivo and in vitro, and neuroprotective effects against neurodegenerative conditions both in vivo and in vitro [9].

Uvaias, similar to other fruits, display a metabolic chemical profile that is characterized by compounds linked to their nutritional value, aroma, flavor, and beneficial health properties. These compounds can consist in ubiquitous primary metabolites that are involved in basic functions, such as amino acids, sugars, organic acids, or secondary metabolites that are fruit-specific, such as phenolics. Recent work has addressed the quality of this fruit [4,6,8,10,11,12]; however, there are still unreported characteristics. Together, these compounds can be applied as potential quality, origin, and authenticity markers in fruits and fruit-derived foods [13].

Conventional physicochemical characterization techniques are routinely applied to define fruit quality, although they require numerous determination steps, in addition to specific equipment, time, and specialized labor [14]. Thus, analytical techniques for multicomponent assessments become interesting, allowing for the simultaneous verification of different parameters. In this context, nuclear magnetic resonance (NMR) spectroscopy has been widely employed in the description of metabolic fruit profiles, assisting in determining the fruit composition [1,15,16]. The preparation of NMR samples is relatively simple, and the analyses allow qualitative and quantitative evaluations concerning a wide range of metabolites in complex mixtures, similar to those that are present in biological samples.

In this scenario, this research assessed uvaia quality attributes by conventional physicochemical parameters and furthered metabolite knowledge in selected uvaia accessions by nuclear magnetic resonance (NMR) spectroscopy. This study comprises the largest number of accessions of uvaia plants that has been investigated to date. It brings the most complete approach to the content of sugar and related compounds, along with acids and their derivatives. In addition, it is unique in providing the first report on the amino acid composition of uvaias.

## 2. Materials and Methods

Uvaias were subjected to quality analysis using physicochemical and metabolite profile analysis, as shown in Figure 1.

### 2.1. Physicochemical Characterization and Accession Selection

#### Plant Material

Uvaias belonging to 41 accessions obtained from the 2020 harvest were sampled in the municipality of Cabo Verde, MG, southeastern Minas Gerais (21°28′19′′ S 46°23′25′′ O). Based on the Köppen climate classification, this region qualifies as Cwa (Cwa: C = mild temperate, w = dry winter, and a = warm summer)—a humid subtropical climate characterized by hot and humid summers and cool to mild winters

Fruits were collected from the accessions identified with the following codes: UV016, UV022, UV027, UV028, UV029, UV030, UV036, UV037, UV039, UV040, UV041, UV042, UV043, UV046, UV047, UV048, UV049, UV050, UV052, UV056, UV057, UV058, UV059, UV060, UV062, UV067, UV068, UV069, UV072, UV073, UV075, UV076, UV088, UB089, UV112, UV114, UV116, UV120, UV122, UV125, and UV146.

The fruit pulp was subjected to the following physicochemical analysis:

Fresh mass: The fresh mass of whole fruits was determined using an analytical balance and the results were expressed in grams (g).

Diameter and height: Both parameters were measured with a digital caliper, and the results were expressed in millimeters (mm).

Yield: Pulp content in relation to the seeds and skin (%).

Seed mass: Pulp and seeds were separated and the mass of the seeds was measured using an analytical balance, with the results expressed in grams (g).

Total soluble solid (TSS): The samples were collected, filtered with gauze, homogenized, and then analyzed by direct reading using a digital Atago PR-101 refractometer (Palette), with the results expressed in °Brix [17].

Titratable acidity (TA): TA was determined by neutralization titration, according to Carvalho et al. [18]. Fresh fruit mass extracts (5 g) were homogenized in 45 mL of distilled water. The extracts were filtered through filter paper and titrated with standard 1 N sodium hydroxide (NaOH) solution until they reached pH 8.1. The results were expressed as % citric acid equivalents in the fresh mass.

After the physicochemical characterizations, ten accessions (UV016, UV030, UV036, UV028, UV037, UV052, UV068, UV088, UV114, and UV125) were selected in order to determine the diversity of quality by NMR. The selection was performed by multivariate analysis, as described in Section 2.2, using the distribution of samples in different quadrants and by sample availability.

These accessions were termed as accessions 1 to 10.

### 2.2. Metabolite Identification and Quantification in Selected Uvaia Accessions

#### 2.2.1. Sample Preparation and Metabolite Extraction

The fruit pulps from each accession were processed and centrifuged and the supernatants were separated. A total of 450 µL of the extracts from each sample was diluted in 450 µL of phosphate buffer (PBS; 0.10 M, pD = 7.3) prepared with deuterated water, followed by the addition of 10 µL of an internal standard 3-trimethylsilyl-2,2,3,3-d4-propionate solution at 0.01 g mL^−1^ (TMSP in D_2_O). The pH of each sample was adjusted to 3.75 ± 0.01, with NaOH (1 mol L^−1^) or HCl (1 mol L^−1^), and 600 µL of each sample was then transferred to 5.0 mm NMR tubes for the NMR analyses.

#### 2.2.2. Metabolomics Analyses

The NMR spectra were obtained at 25 °C by employing a 14.1 Tesla Bruker AVANCE III (600 MHz for hydrogen frequency) equipped with a 5 mm PABBO (Broad Band Observe) direct detection probe with ATMA^®^ (Automatic Tunning Matching Adjustment), a z-field gradient, BCU-I variable temperature unit, field gradient generator unit, and a Sample-Xpress™ automatic sample changer. The ^1^H NMR spectra were acquired through a NOESY 1D pulse sequence (named noesygppr1d in the TopSpin Bruker software) with a field gradient and water signal suppression by irradiation at 2820.39 Hz (O1). The equipment conditions were as follows: 128 measurements (ns), 4 dummy scans (ds), 65536 data points during acquisition (td), spectral window of 20.0276 ppm (sw), fixed receiver gain (80.6 rg), calibrated 90° pulse (13.821 µs), acquisition time between each acquisition of 2.76 s (aq), waiting time between each averaging 5*T1 (19 s), and a mixing time of 5 ms (d8). The ^1^H NMR spectra were referenced through the TMSP-d4 signal at 0.0 ppm.

The ^1^H spectra were assigned and quantified using the Chenomix NMR software (Chenomix Inc, Edmonton, Canada), in which the overlapping signals of each compound were deconvolved based on the database (library) of the Chenomix NMR software itself. The compound identification, when necessary, was confirmed by means of 2D NMR experiments ^1^H Jres, 1H-^13^C HSQC, and ^1^H-^1^H COSY, performed on selected samples. The metabolite quantification was performed in relation to the TSPd4 standard concentration.

#### 2.2.3. Statistical Analyses

The physicochemical uvaia quality data are presented as averages and coefficients of variation and are represented as a heatmap chart. The metabolite quantification data were subjected to an analysis of variance and classification of means according to Tukey’s test and to a principal component analysis. Multivariate analyses were performed using the Matlab 2010a software. The classification of selected uvaia accessions was performed according to metabolic profile. The Milano PCA toolbox and the Classification toolbox were used for the discriminant analyses. A pictorial representation of uvaia metabolism was prepared based on metabolic maps available in the Kyoto Encyclopedia of Genes and Genomes (KEGG) [19] from a metabolic pathway analysis performed using the pathways previously identified in the KEGG of *Arabidopsis thaliana* (thale cress). For this purpose, the MetaboAnalyst 5.0 platform was also used.

## 3. Results

### 3.1. Physicochemical Uvaia Characterization and Accession Selection

The results from several accessions showed an average height of between 16.21 ± 0.16 mm (UV016) and 32.62 ± 0.80 mm (UV068) (Table 1). The diameter was evaluated between 19.51 ± 0.19 mm (UV036) and 41.17 ± 0.74 mm (UV068). The verified fresh weight ranged from 4.29 ± 0.27 g (UV036) to 27.90 ± 1.61 (UV068), while the seed mass ranged from 0.84 ± 0.16 (UV114) up to 5.47 ± 1.45 (UV068). The yield values were between 61.95 ± 4.94 (UV088) and 89.95 ± 1.19 (UV030). The soluble solids content varied from 5.87 ± 0.21 (UV088) to 15.30 ± 0.46 °Brix (UV028). The total titratable acidity content was found to fluctuate between 0.83 ± 0.39% citric acid eq. (UV052) and 3.70 ± 0.06% citric acid eq. The standard deviations are available in the Appendix A.

Ten uvaia accessions were selected according to the analyzed variables, the sample quadrant distribution, and the sample availability, marked in red in Figure 2.

### 3.2. Metabolite Identification and Quantification in Selected Uvaia Accessions

The pulps were extracted from the fruits belonging to accessions UV016, UV030, UV036, UV028, UV037, UV052, UV068, UV088, UV114, and UV125, termed accessions 1 to 10, for the NMR analysis.

Twenty-six different chemical compounds were identified in the assigned spectra that were obtained from the uvaia supernatants (Figure 3), comprising predominantly organic acids, sugars, and amino acids. For details of the 2D structures of all of the identified compounds, refer to Appendix A.

The sugars glucose and fructose, and the organic acid malate, were the most noteworthy among the identified compounds, which were all present at higher concentrations compared to the other identified components (Table 2 and Table 3).

Thirteen amino acids and related compounds were identified in uvaias (Table 4). Stacked bar graphs containing the percentages of sugars, acids, amino acids, and related compounds are shown in Appendix A, respectively.

The uvaia sugar and organic acids ratios (S/OA) varied between 8.91 and 17.95 (Table 5), with 0.69 correlating with the SSC/TA ratio, in which 8.91 (S/OA) corresponds to 2.78 (SSC/TA) and 17.95 (S/OA) corresponds to 13.92 (SSC/TA).

### 3.3. Selected Uvaia Accession Classification

A principal component analysis (PCA) and a partial least squares discriminant analysis (PLS-DA) were performed when analyzing the metabolite concentrations of each uvaia accession that was selected in Figure 2. The PCA models indicate natural variable groupings and are adjusted for each class based on the direction that demonstrates the greatest dataset variation [20]. Each sample that is depicted in Figure 4 represents a repetition of the studied accessions in the PLS-DA analysis. The loadings of latent variables one and two followed the same pattern as the PCA loadings (Appendix A).

Figure 5 indicates three main classifications. Class 1 includes accessions two and four, demonstrating that TSS and Y are the main parameters of this class and that they are probably influenced by most of the metabolites that are found in this fruit, with fructose, glucose, and ethanol being relevant for TSS and gallate and methanol being relevant for Y. Class 2 was determined by the measurement parameters (N seed, S, FW, D, and H) and comprises accessions seven and eight, with threonine, glutamine, phenylalanine, and valine consisting in the most relevant metabolites for this class. The third and last class comprises all of the other accessions (3, 5, 6, 9, and 10) and is distinguished by TTA with leucine, isoleucine, and cinnamate, with these compounds comprising the relevant metabolites for this class. The metabolites that were responsible for each class were confirmed with heatmaps (Appendix A). Accession one was excluded in this analysis since, instead of triplicates for the physicochemical analysis, only a duplicate was available, with weak data being a correlation with metabolomics data.

The metabolites that have been identified in uvaias are depicted in Figure 6, highlighting the metabolisms to which they belong through the pathway analysis that was carried out considering the uvaia classifications according to their physicochemical characteristics (Appendix A).

## 4. Discussion

The quality characteristics shown among the 41 accessions that were evaluated (Table 1) can offer diversified industrial uses and sensory experiences delivered by these fruits. For instance, the fruits with a higher yield (UV030) may be interesting for pulp production, while sweeter fruits (UV028) may provide a better experience for fresh consumption. The fruits that have been studied here have presented a greater range of values in the examined parameters. One example refers to the soluble solids content, which shows larger variation than that previously reported [7,21]. Furthermore, the assessed titratable acidity value is more extensive than that previously described [7].

Ten of the most widely diverse accessions were selected to be subjected to metabolomics study through NMR analysis, which obtained spectra such as the one depicted in Figure 2. The uvaia spectra obtained here are in agreement with other plant metabolome assessments, in which the presence of amino acids is generally attributed to the 0–3 ppm region, while sugars appear in the 3–6 ppm region, and phenolic compounds from 6 ppm onwards [22].

Sucrose, glucose, and fructose were the sugars identified, which is in agreement with that previously studied in six uvaia accessions [4]. Glucose was the predominant sugar in 9 of the 10 analyzed accessions, except for accession 9, where the highest concentration sugar was fructose (Table 2). Sucrose was the third most quantified sugar in uvaias. Comparing the sucrose concentrations, a non-reducing sugar, to glucose and fructose concentrations, uvaias display a ripening pattern that is similar to other Myrtaceae fruits, such as *Myrcianthes pungens* and java plums *(Syzygium cumini*) [23]. Reducing sugars are predominant in these fruits, in contrast to what has been reported for cambucis (*Campomanesia phaea*), another Myrtaceae fruit, where sucrose is noteworthy even in ripe fruits. It is common for uvaias to present an acidic sweet taste, corroborated by the high sugar expression levels that have been observed in the samples [4,24].

As a highly perishable fruit, the presence of ethanol and methanol among the metabolic profile in uvaias was expected (Table 2). Anaerobic respiration during the postharvest period favors ethanol synthesis through the anaerobic respiratory pathway by acetaldehyde reduction after glycolysis [25]. Ethanol concentrations in the evaluated accessions ranged from 18.3 mg 100 g^−1^ to 65.7 mg 100 g^−1^, averaging 30 mg 100 g^−1^, which is above the values that have been reported for cambucis [1]. Alongside another 59 compounds, ethanol contributes to the a volatile compound profile, identified in the mature fruit stage [7]. This compound has also been observed in jaboticabas (*Plinia* sp.), another Myrtaceae fruit [26]. The ethanol that is present in the fruits influences consumer sensory experiences, altering the flavor and aroma perceptions. Fruits containing excessive ethanol are associated to over-ripeness.

Herein, uvaia methanol content varied between 6.6 mg 100 g^−1^ and 24 mg 100 g^−1^, formed mainly from the demethylation of certain macromolecules, such as cell wall pectin, due to the action of the pectinmethylesterase (PME) [27] (Table 2). Methanol synthesis links to fruit ripening, accompanying the natural breakdown that is derived from cell wall softening, where about 40% of the pectin-bound methanol is released in this process [28]. The lethal amount of orally ingested methanol for humans ranges from 0.3 to 1 g per kg^−1^ body weight [29]. As the maximum concentration of methanol that was observed in the analyzed uvaia extracts was of 24 mg 100 g^−1^ (Table 2), which is similar to that of tomato juice diluted by half [30], even a child weighing only about 10 kg would have to consume several dozen kg of uvaia to ingest a lethal dose. Similarly to ethanol, methanol belongs to the set of volatile compounds that are emitted in fruits such as apples and pears, fruit juices, and vegetable juices [28,30]. In this regard, Li et al. [31] detected the presence of methanol, not only in fruits (apples and peaches), but also in fruit leaves, during the four stages of development, with methanol being responsible for the emission of 28.8% of all of the analyzed volatile compounds in apple tree leaves, 13.9% in apples, 41.6% in peach leaves, and 27.1% in peaches. No significant differences in methanol emissions during apple and peach ripening stages were observed, revealing a common methanol occurrence [31].

The uridine content in uvaia accessions varied between 0.62 mg 100 g^−1^ and 2.12 mg 100 g^−1^ (Table 2). Uridine, a nucleoside originated from glucose-6-phosphate, often associates with sugars, called UDP-sugars, in plants. Nucleotide sugars lead to all glycosylation reactions and are necessary for the synthesis of oligo- and polysaccharides and for protein and lipid glycosylation of [32]. Among all nucleotide sugars, UDP-sugars participate in biomass production, as well as in the synthesis of cellulose, hemicellulose, and pectins that are used in cell wall production [32].

The organic acid (malate, gallate, citrate, and succinate) contents are in agreement with those recently reported in the literature [24] (Table 3). Malic acid comprises the most present fraction among the acids that have been identified, followed by citric acid. The literature reports indicate malic acid levels in uvaias ranging from 189 mg 100 g^−1^ to 628 mg 100 g^−1^, and the levels determined herein ranged between 142 mg 100 g^−1^ and 738 mg 100 g^−1^. Malic acid, a dicarboxylic acid derived from succinic acid, has a mild sour taste that persists in the mouth without transmitting an overly potent flavor [33]. The presence of malic acid surpassed citric acid between 4.9-fold (uvaia accession one) and 11.3-fold (uvaia accession four), potentially leading to different sensory experiences. The acid accumulation in fruit vacuoles depends on the genotype, with citrus fruits containing more citric acid, and fruits such as apples and pears presenting a higher malic acid content [34]. The malic acid resulted from the tricarboxylic acid cycle, which is the main metabolic pathway for energy production in plant cells. The presence of tricarboxylic acid (TCA) intermediates, e.g., malate and succinate, are related to GABA, whose metabolization of the latter goes through the TCA pathway stages [35].

This study presents the first report of GABA content in uvaias, which was from 0.520 mg 100 g^−1^ to 3.611 mg 100 g^−1^ (Table 3). In addition to its functions as a non-protein amino acid in plants, GABA acts as a potent bioactive compound in humans. A recent exploration of bioactive molecules from natural sources for adoption in diets is noted, as well as a search for the fortification of horticultural products with GABA for health purposes. Concentrations between 16 and 61 mg 100 g^−1^ of GABA have been reported in 22 potato varieties [36].

Gallic acid plays a part in the phenolic compounds that are present in this fruit, with values ranging from 0.36 mg 100 g^−1^ to 0.66 mg 100 g^−1^ (Table 3). The content of total phenolic compounds has been previously described for uvaias, with an average value of 119 mg 100 g^−1^ [6]. Gallic acid can be found in its free form or as part of tannins (called galotannin), which in recent studies have been reported to have health benefits, such as antioxidant, anticoagulant, antiallergic, anti-inflammatory, anthelmintic, and antimicrobial activities [37]. Both tannins and gallotannins classify as bioactive molecules. Gallotannins derive from the union of gallic acid and glucose, considered hydrolyzable tannins, presenting beneficial health effects such as antioxidant, anti-inflammatory, and antidiabetic activities [38].

Cinnamic and formic acids were also identified in uvaia pulp (Table 3). The presence of cinnamic acid in uvaias varied more than fourfold between the fruits with lower and higher concentrations. Following hydroxylation, cinnamic acid produces p-hydroxy cinnamic acid or *p*-coumarate, which is a plant monophenol precursor to the production of various di(lignans) and polyphenols (lignins) [39]. In turn, small amounts of formic acid can be found in fruits, probably playing a role in the defense against phytopathogens [32]. Acetamide may be involved in coordinating the exchange between growth and stress resistance, linked to the synthesis of auxin and abscisic acid [40].

This report comprises the first amino acid profile description from this fruit (Table 4). Essential amino acids are not synthesized by humans, and therefore must be acquired externally. Isoleucine, leucine, phenylalanine, threonine, and valine are present in uvaias, and are classified as essential for humans [41]. According to Dietary Reference Intakes (DRI), the reported values of dietary requirements for essential amino acids in healthy adults are as follows: Ile, 19; Leu, 42; Phe + Tyr, 33; Thr, 20; and Val, 24 [mg kg^−1^ d^−1^) [42,43]. Arginine, glutamine, proline, and tyrosine are classified as conditionally indispensable, while alanine, aspartate, and serine are classified as dispensable.

Glutamine and alanine were the amino acids that were present in higher amounts in uvaia, ranging from 10.62 mg 100 g^−1^ (accession two) to 78.65 mg 100 g^−1^ (accession eight) for glutamine and between 4.17 mg 100 g^−1^ (accession ten) and 10.90 (accession two) for alanine (Table 4). Glutamine has been indicated as the most abundant amino acid in pistachios, wheat flour, and white rice, and the second most abundant amino acid in corn, peanuts, potatoes, and soybeans [44]. Several amino acids taste sweet or attractive (such as umami) to humans and are also interesting to rodents and other animals [45]. Flavor descriptors such as “sweet” are attributed to the amino acids alanine, glutamine, phenylalanine, proline, serine, and threonine, which were present in the evaluated uvaia accessions.

The amino acids in uvaia closely correlate with volatile compound emission, affecting its aroma, especially in fully ripened fruit. At this ripening stage, compounds such as esters and fatty acid derivatives are produced in high amounts, due to the degradation of the cell membranes and the amino acids in the reactions that are catalyzed by aminotransferases [7]. Alanine, valine, leucine, and isoleucine serve as precursors for aldehydes, alcohols, esters, acids, and nitrogen and sulfur-containing volatiles, and phenylalanine originates from phenylpropanoids/benzenoids [46]. Uvaias release a highly attractive aroma, presenting citrus, fruity, and sweet notes when they are ripe. The lowest coefficient of variation for amino acids amounted to 21.96% for threonine and the highest value corresponded to 55.80% for glutamine, indicating variable levels in the distinct accessions, potentially directly affecting the flavor and the aroma.

The metabolite coefficients of variation were always above 21%, reaching levels greater than 85% (sucrose) (Table 2, Table 3 and Table 4). This lack of quality attribute uniformity reflects uvaia accessions’ diversity, as this fruit tree has not yet been subjected to genetic improvement programs, revealing uvaias’ potential to serve as a raw material for a wide spectrum of interests, such as in natura consumption, processed consumption, and use in the development of foods presenting bioactive and pharmaceutical properties.

Based on the values that were obtained from the quality parameters, here we suggest an index for fruit evaluation. The S/OA ratio could replace the SS/TA ratio to determine quality and could predict uvaia SS/TA ratio values (Table 5). Additionally, after the NMR evaluation, a tentative grouping of samples was attempted. Generally, the application of ^1^H NMR in plants is more complex than the study of biofluids, due to the rich metabolite information. Because of this, chemometric analyses are commonly used in metabolomics assays for the discrimination of samples based on their spectral information [47].

A grouping was noted between the samples and, despite the closeness of accessions three, five, and six (Appendix A), these groups could still be distinguished. The groups that shifted to the negative axis of PC1 are more influenced by cinnamate, phenylalanine, glutamine, and leucine concentrations. This may indicate a greater exacerbation of the phenylpropanoid pathway. Accession four, in turn, seems to be the most influenced by the positive loadings in PC1, with sugars (fructose, glucose, and sucrose) as the most noteworthy. In absolute terms, the sugar concentrations were higher in accession four and, despite sample self-scaling, these values still influenced the scores of such samples the most. The model presented an explained variance of 92.2%.

When applying the discriminant PLS-DA analysis, the clustering trend was maintained (Figure 4), with an explained variance of 89% and a validation accuracy rate of 0.03. The PLS-DA allows for the elimination of multicollinearity in a set of variables, in this case the spectrum regions that contain information on compound concentrations, from a regression model. This leads to a reduction in the initial data set size, in such a way that the resulting subset of descriptive variables is ideal for predicting the dependent variable, comprising the class variable.

The metabolites that were identified in the uvaias were distributed in nine plant metabolic pathways. The metabolic analysis of the accession’s class allowed the observation of the regulation and accumulation of phytochemicals, depending on each plant. Therefore, according to the characteristics of each accession’s class, the significance of each pathway was shown. Since the uvaia is a non-domesticated fruit, these findings may contribute to the development of quality standards for it [48].

Despite uvaia’s classification having presented different relevant metabolic pathways, the results represent a common metabolism in uvaia plants. Uvaia Classification 1 (TSS, Y) indicated changes in the following metabolisms: isoquinoline alkaloid biosynthesis; alanine, aspartate, and glutamate metabolism; the citrate cycle (TCA cycle); arginine and proline metabolism; pyruvate metabolism; tyrosine metabolism; starch and sucrose metabolism; arginine biosynthesis; and the glyoxylate and dicarboxylate metabolism (Figure 6, Appendix A). Uvaia Classification 2 (S, FW, H) revealed changes in the phenylalanine metabolism; alanine, aspartate, and glutamate metabolism; glycine, serine, and threonine metabolism; arginine biosynthesis; and the glyoxylate and dicarboxylate metabolism. While uvaia Classification 3 (TA) presented alterations in the following metabolisms: isoquinoline alkaloid biosynthesis; alanine, aspartate, and glutamate metabolism; the citrate cycle (TCA cycle); pyruvate metabolism; glycine, serine, and threonine metabolism; tyrosine metabolism, arginine biosynthesis; and the glyoxylate and dicarboxylate metabolism.

Identifying the most active metabolic pathways in each accession class, and their close relationship with phenotypic characteristics, may help in breeding programs for these fruits in the future. For instance, in order to modify the content of soluble solids and yield, changes in the most active pathways that were described in Classification 1 could be considered.

The identification of the metabolic pathways that are active in uvaias (Figure 6) allows comparison with other fruits and provides insight into metabolites that have not yet been identified—probable precursors of identified metabolites. It also connects to the profile of volatile compounds that have already been reported for uvaias, as the breakdown of several of these elements releases aroma compounds [7].

## 5. Conclusions

Twenty-six uvaia metabolites were identified through nuclear magnetic resonance. Six metabolites comprise sugars and related compounds, eight are acids and related compounds, and thirteen consist of amino acids and related compounds. This is the most comprehensive report on sugars and acids in uvaias. The most abundant metabolites were the reducing sugars glucose and fructose, as well as malic and citric acids. This is also the first description of the amino acid uvaia profile.

High variability was observed concerning the evaluated attributes that were intrinsically associated to the biodiversity of this fruit tree. For future plants to be selected in breeding programs, the quality range description work that has been performed here is essential. In this instance, the results are considered to be supporting data for plant pre-breeding. This variability makes it possible to envision potential applications, such as in the food and pharmaceutical fields. The principal component analysis and the partial least squares discriminant analysis results reinforce the differences between the analyzed uvaia accessions.

Identifying the metabolites of uvaia has made it possible to outline for the first time the metabolic pathways of this fruit.

## Figures and Tables

**Figure 1 foods-12-01881-f001:**
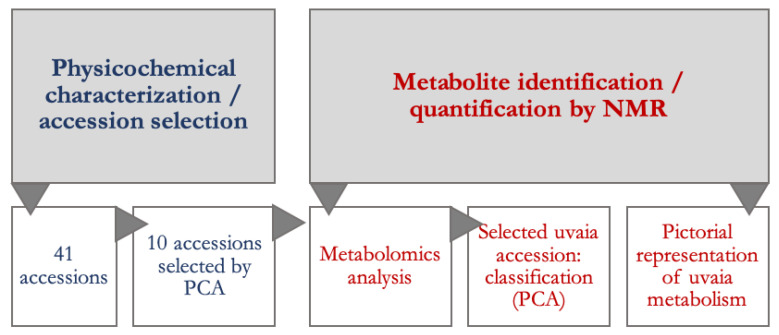
Procedure flow for quality analysis of uvaias.

**Figure 2 foods-12-01881-f002:**
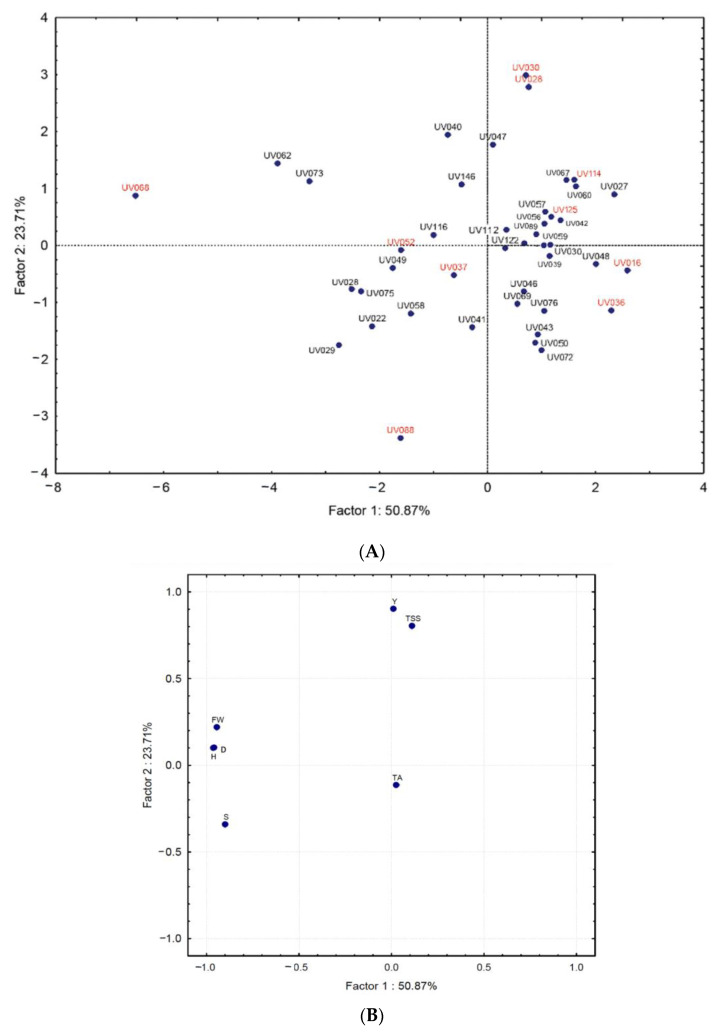
Principal component analysis (**A**) scores and (**B**) loadings regarding uvaia quality attributes. Y: yield; TA: titratable acidity; FW: fresh weight; S: seed mass; D: diameter; H: height; and TSS: total soluble solids.

**Figure 3 foods-12-01881-f003:**
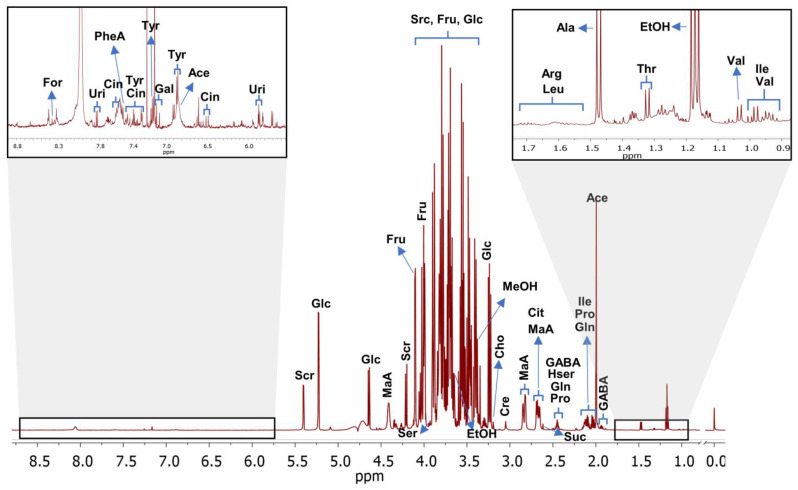
^1^H NMR uvaia supernatant spectra. Compound identified in uvaia pulp supernatants. Formate (For, 1); uridine (Uri, 2); cinnamate (Cin, 3); acetamide (Ace, 4); phenylalanine (PheA, 5); tyrosine (Tyr, 6); gallic acid (Gal, 7); sucrose (Scr, 8); glucose (Glc, 9); fructose (Fru, 10); malic acid (MaA, 11); serine (Ser, 12); methanol (MeOH, 13); hill (Cho, 14); ethanol (Eth, 15); glutamine (Gln, 16); citrate (Cit, 17); succinate (Suc, 18); proline (Pro, 19); isoleucine (Ile, 20); 4-aminobutyric (GABA, 21); arginine (Arg, 22); leucine (Leu, 23); alanine (Ala, 24); threonine (Thr, 25); and valine (Val, 26). Reference: Sodium-d4 3-trimethylsilylpropionate (TMSP).

**Figure 4 foods-12-01881-f004:**
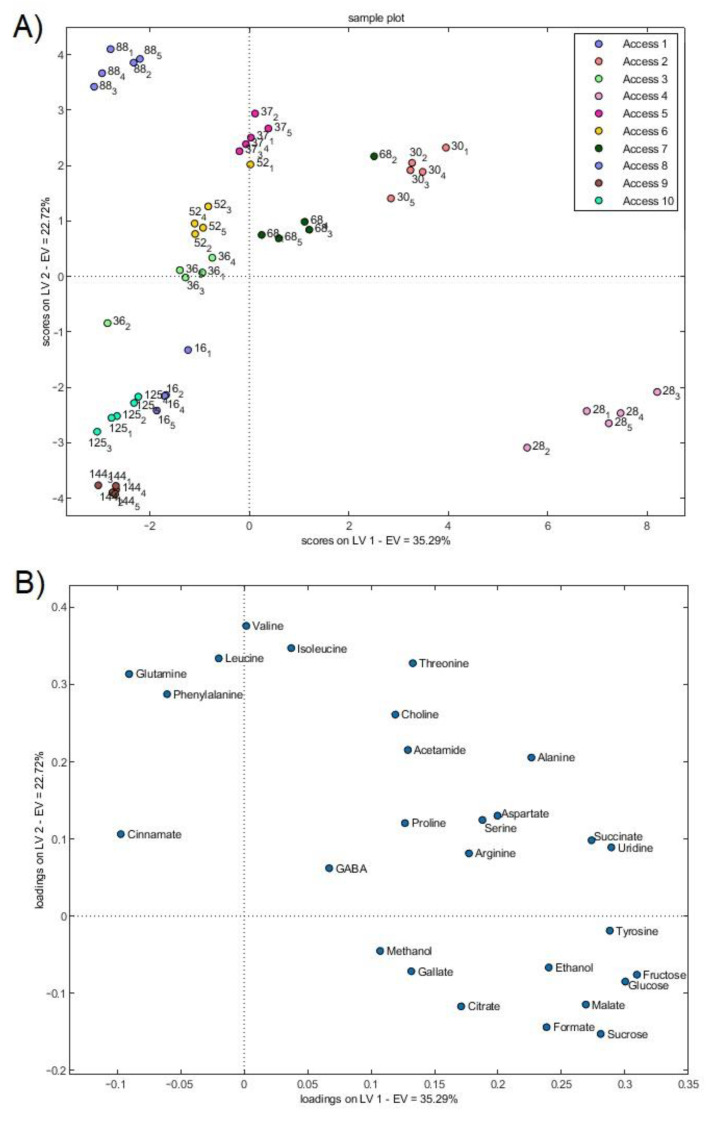
(**A**) Partial least squares discriminant analysis (PLS-DA) scores and (**B**) loadings of uvaia quality attributes according to the metabolites identified by NMR.

**Figure 5 foods-12-01881-f005:**
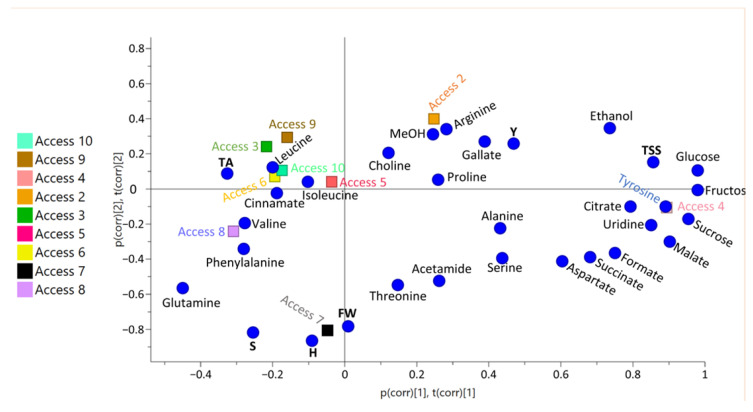
PCA correlation loadings concerning uvaia quality attributes according to physicochemical data and metabolic profile data. H: height (mm), FW: fresh weight (g), S: seeds mass (g), Y: yield (%), TSS: total soluble solids (°Brix); TA: titratable acidity (% citric acid eq).

**Figure 6 foods-12-01881-f006:**
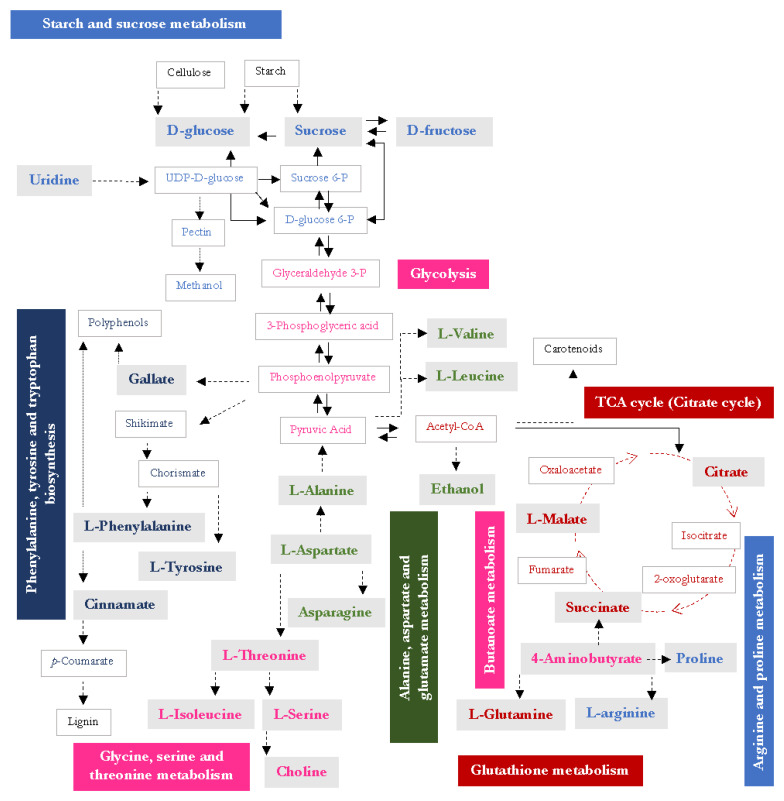
Pictorial representation of uvaia metabolism. The metabolites found in this study are highlighted with a gray background. UDP: Uridine-D-Glucose; P: phosphate.

**Table 1 foods-12-01881-t001:** Physicochemical uvaia characteristics—2020 harvest, including heatmap distribution analysis. Increased and decreased values are given in red and blue, respectively. See Appendix A for accession standard deviations.

	H	D	FW	S	Y	TSS	TA
UV016	16.21	20.83	4.80	1.26	74.13	10.08	1.69
UV022	24.83	30.22	13.85	4.26	70.91	8.12	1.48
UV027	17.96	21.11	5.41	1.19	78.30	12.73	2.08
UV028	22.08	26.18	8.91	1.34	85.07	15.30	1.39
UV029	26.38	33.82	14.18	3.46	75.94	7.52	1.25
UV030	20.67	25.09	12.88	1.24	89.95	13.92	1.00
UV036	19.20	19.51	4.29	1.26	70.79	8.73	1.10
UV037	24.11	28.03	10.28	2.48	75.92	8.40	1.06
UV039	20.30	25.51	6.83	1.10	84.04	6.57	0.95
UV040	24.99	30.36	11.64	1.73	85.58	12.07	1.29
UV041	23.10	25.80	9.30	2.89	69.24	8.42	1.36
UV042	19.92	24.05	6.51	1.15	82.53	9.10	1.22
UV043	20.92	25.50	6.65	1.48	64.87	8.72	1.10
UV046	21.94	23.69	7.44	2.26	70.72	10.02	2.02
UV047	23.82	27.04	9.92	1.56	84.06	12.30	1.11
UV048	18.59	21.65	5.55	1.40	74.60	10.27	2.31
UV049	26.16	30.21	13.14	3.18	76.35	8.95	2.26
UV050	20.49	23.67	6.54	2.15	65.95	8.62	1.47
UV052	25.56	31.99	12.92	2.42	81.39	7.28	0.83
UV056	21.50	24.65	7.75	1.50	80.76	9.12	1.47
UV057	20.19	26.24	7.36	1.14	84.70	8.13	1.18
UV058	24.97	28.03	12.46	3.45	73.51	7.58	1.27
UV059	20.09	24.90	7.46	1.53	79.65	8.98	1.38
UV060	20.00	23.19	6.81	1.15	83.10	11.38	2.07
UV062	28.17	32.03	26.23	3.96	85.60	10.43	1.69
UV067	19.81	23.88	7.61	1.06	86.20	10.20	1.49
UV068	32.62	41.17	27.90	5.47	80.47	10.20	1.03
UV069	21.39	24.00	8.08	2.58	69.27	10.45	3.48
UV072	19.10	24.97	6.59	1.82	72.99	5.98	1.77
UV073	30.09	34.94	18.19	3.02	83.45	10.53	3.46
UV075	26.74	31.25	13.63	4.05	70.33	10.38	3.13
UV076	20.06	24.59	6.79	1.91	72.49	8.77	3.58
UV088	24.05	28.82	11.12	4.24	61.95	5.87	3.17
UV089	21.40	26.93	7.40	1.54	79.37	9.50	3.08
UV112	22.11	26.91	9.22	1.60	82.65	8.92	2.85
UV114	20.03	24.39	6.64	0.84	87.59	10.17	3.16
UV116	22.79	29.55	13.81	2.72	80.37	9.55	3.01
UV120	20.50	24.19	7.45	1.47	80.79	9.05	2.87
UV122	22.42	24.91	9.37	2.08	77.09	9.78	1.65
UV125	21.40	24.24	6.85	1.26	81.58	10.28	3.70
UV146	25.62	28.14	10.49	1.81	82.59	11.17	3.13

Accession identification codes: UV = uvaia; three-digit number = accession identification. H: height (mm), D: diameter (mm), FW: fresh weight (g), S: seed mass (g), Y: yield (%), TSS: total soluble solids (°Brix), and TA: titratable acidity (% citric acid eq).

**Table 2 foods-12-01881-t002:** Sugars and related compounds quantified by NMR and expressed as mg 100 g^−1^.

Accession	Sucrose		Glucose		Fructose		Ethanol		Methanol		Uridine	
1	302.6	b	1800.9	b	1643.1	b	23.5	b	10.8	b	1.27	ab
2	567.1	b	2832.3	ab	2832.1	ab	65.7	a	24.0	a	1.40	ab
3	303.2	b	1285.6	b	1258.5	b	23.8	b	10.7	b	0.93	b
4	1999.5	a	5203.6	a	4279.8	a	51.6	ab	10.1	b	2.12	a
5	405.0	b	2022.3	b	1819.6	b	18.3	b	9.9	b	1.59	ab
6	193.9	b	1739.4	b	1660.7	b	28.1	b	14.1	ab	1.17	ab
7	569.8	b	1970.3	b	1752.4	b	22.4	b	13.5	ab	1.36	ab
8	600.9	b	2044.9	b	1639.2	b	21.2	b	6.6	b	1.44	ab
9	253.1	b	1045.0	b	1178.0	b	25.7	b	11.8	b	0.81	b
10	360.3	b	1398.8	b	1294.0	b	20.0	b	13.4	b	0.62	b
Mean	555.5		2134.3		1935.7		30.0		12.5		1.27	
CV (%)	85.28		50.22		44.78		49.30		36.92		30.26	

The different lowercase letters in the columns indicate statistical difference among accessions according to Tukey’s test (*p* < 0.05).

**Table 3 foods-12-01881-t003:** Organic acids and related compounds quantified by NMR and expressed as mg 100 g^−1^.

Accession	Citrate		Malate		Gallate		Succinate		4-Aminobutyrate		Cinnamate		Formate		Acetamide	
1	49.41	a	241.95	b	0.52	a	0.038	ab	1.392	b	0.79	b	0.113	b	107.44	abc
2	33.22	b	312.14	b	0.47	a	0.065	ab	0.520	b	0.58	b	0.120	b	88.11	abc
3	23.97	b	250.54	b	0.63	a	0.041	ab	1.379	b	0.68	b	0.062	b	97.35	abc
4	65.27	ab	737.97	a	0.66	a	0.079	a	0.887	b	0.72	b	0.205	a	109.36	abc
5	36.68	ab	299.35	b	0.50	a	0.061	ab	2.251	ab	0.48	b	0.075	b	138.87	a
6	21.61	b	196.79	b	0.58	a	0.032	ab	1.090	b	0.39	b	0.111	b	72.53	bc
7	33.11	b	296.04	b	0.37	a	0.040	ab	1.922	ab	0.31	b	0.130	b	120.057	abc
8	41.38	ab	369.61	b	0.36	a	0.063	ab	3.611	a	1.43	a	0.166	a	106.37	abc
9	27.03	b	141.53	b	0.47	a	0.034	ab	1.437	b	0.85	ab	0.060	b	76.39	bc
10	21.54	b	318.96	b	0.43	a	0.019	b	0.999	b	0.55	b	0.071	b	56.92	c
Mean	35.32		316.49		0.50		0.05		1.55		0.68		0.10		97.34	
CV (%)	36.29		47.91		19.83		35.68		53.90		46.84		38.60		23.20	

The different lowercase letters in the columns indicate statistical difference among accessions according to Tukey’s test (*p* < 0.05).

**Table 4 foods-12-01881-t004:** Amino acids and related compounds quantified by NMR and expressed as mg 100 g^−1^.

Accession	Ala		Arg		Asp		Cho		Gln		Ile		Leu		Phe		Pro		Ser		Thr		Tyr		Val	
1	8.32	abc	2.57	ab	4.56	ab	1.53	b	37.36	ab	0.64	ab	0.82	abc	0.86	b	1.89	a	3.84	a	2.44	ab	2.00	b	1.50	ab
2	10.90	a	3.37	a	6.46	ab	3.34	a	10.62	b	0.88	a	0.97	a	0.91	b	1.95	a	4.61	a	2.57	ab	2.96	b	1.34	ab
3	7.88	abc	0.97	b	6.90	ab	1.60	b	36.47	ab	0.49	b	0.53	bc	0.83	b	1.07	ab	3.11	a	2.59	ab	1.80	b	1.13	ab
4	8.44	abc	2.26	ab	8.42	a	1.06	b	14.06	b	0.38	b	0.49	bc	0.62	b	1.30	ab	4.14	a	2.23	ab	5.47	a	0.73	b
5	8.95	abc	1.96	ab	5.29	ab	1.88	ab	51.12	ab	0.62	ab	0.75	abc	1.00	b	1.28	ab	3.82	a	2.99	ab	1.51	b	1.73	a
6	10.30	ab	2.30	ab	3.96	b	2.22	ab	36.15	ab	0.88	a	0.85	ab	0.63	b	1.73	ab	4.28	a	2.80	ab	2.89	b	1.52	ab
7	9.14	abc	1.56	ab	5.10	ab	1.69	b	45.24	a	0.55	ab	0.58	bc	0.95	b	1.00	ab	4.87	a	3.01	a	1.56	b	1.35	ab
8	6.46	abc	0.88	b	8.48	a	1.60	b	78.65	ab	0.76	ab	0.96	a	2.23	a	1.06	ab	3.37	a	3.09	a	3.16	ab	1.64	a
9	5.48	bc	0.97	b	3.26	b	1.38	b	34.91	b	0.47	b	0.62	abc	1.12	b	1.05	ab	2.42	a	2.00	ab	1.44	b	1.04	ab
10	4.17	c	1.01	b	3.35	b	0.89	b	16.31	b	0.38	b	0.53	bc	0.47	b	0.60	b	2.41	a	1.30	b	1.12	b	0.72	b
Mean	8.00		1.79		5.58		1.72		36.09		0.60		0.71		0.96		1.30		3.69		2.50		2.39		1.27	
CV (%)	26.21		47.42		34.50		39.77		55.80		30.80		25.78		50.62		33.70		23.80		21.96		54.23		28.05	

The different lowercase letters in the columns indicate statistical difference among accessions according to Tukey’s test (*p* < 0.05). Ala: alanine; Arg: arginine; Asp: asparagine; Cho: choline; Gln: glutamine; Ile: isoleucine; Leu: leucine; Phe: phenylalanine; Pro: proline; Ser: serine; Thr: threonine; Tyr: tyrosine; and Val: valine.

**Table 5 foods-12-01881-t005:** Means for the different metabolite groups, sugars, organic acids, amino acids, and sugar organic acid ratio (S/OA) determined by NMR and expressed as mg 100^−1^.

Accession	Sugar		Rel. Sugar		Organic Acids		Rel. Organic Acids		Am. Acids		Sum	S/OA
1	3746.60	b	34.33	b	294.21	b	107.44	abc	69.59	b	4252.17	12.73
2	6231.56	ab	89.64	a	347.11	b	88.11	abc	52.28	b	6808.71	17.95
3	2847.35	b	34.46	b	277.31	b	97.35	abc	66.30	b	3322.78	10.27
4	11482.88	a	61.75	ab	805.79	a	109.36	abc	51.70	b	12511.48	14.25
5	4246.93	b	28.22	b	339.39	b	138.88	a	84.51	ab	4837.92	12.51
6	3593.97	b	42.17	b	220.60	b	72.53	bc	71.69	b	4000.97	16.29
7	4292.48	b	35.88	b	331.92	b	120.06	abc	77.97	b	4858.31	12.93
8	4284.93	b	27.80	b	416.52	b	106.37	abc	113.78	a	4949.40	10.29
9	2476.08	b	37.44	b	171.40	b	76.39	bc	56.96	b	2818.27	14.45
10	3053.01	b	33.34	b	342.56	b	56.92	c	33.89	c	3519.73	8.91
Mean	4625.58		42.50		354.68		97.34		67.87		5187.97	13.06
CV%	56.77		45.04		48.79		25.03		32.15		54.03	21.48

The different lowercase letters in the columns indicate statistical difference among accessions according to Tukey’s test (*p* < 0.05).

## Data Availability

Data available on request.

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
