# Peer review of "Quality Attributes and Metabolic Profiles of Uvaia (Eugenia pyriformis), a Native Brazilian Atlantic Forest Fruit"

_foods, 2023, doi:10.3390/foods12091881_

Round 1

Reviewer 1 Report

The study aimed to assess Uvaia quality attributes by conventional physicochemical parameters and furthered metabolite knowledge in selected Uvaia accessions by nuclear magnetic resonance (NMR) spectroscopy. The study described the characteristics of the fruit but did not make any recommendations or importance for it. Please find below my comments for improving the manuscript: 

1- The title needs to use Capitalize Each Word

2- The scientific name of this fruit must be mentioned

3- The abstract needs to be revised. It should clearly explain the purpose of the study and briefly show the most important results obtained and the importance of conducting the study.

4- The introduction is brief It must display the study's importance, novelty, and motivation.

5- The first subsection in Materials and Methods "2.1. Traditional physicochemical uvaia characterization and accession selection" Not clear It should be rephrased, and the physicochemical measurements should be briefly described.

5- The methodology lacks enough details to be fully understood or replicated.

6- I think the statistical analysis symbols displayed in the results tables are not appropriate in these displays. Please add footers for the Tables to describe what is in them.

7- The recommendation and the importance of the study should be added to the conclusions.

Author Response

Please see the file sent as PDF. 

Reviewer 2 Report

The manuscript: Quality attributes and metabolic profiles of uvaia, a native Bra- zilian Atlantic Forest fruit, describes mainly a NMR study of the title plant. It is well documented in most of the sections. In some of the tables and figures it is indicated  the meaning of the abreviate words, as the case of Cho = hill. What do you mean to this?

Author Response

Please see the file sent as PDF. 

Reviewer 3 Report

I have some comments that I hope the author can consider

1. The author needs to highlight the innovation of this manuscript.

2. A comparative analysis of the significant differences between treatments were suggested in table 1.

3. The subtitles of the manuscript should be further enriched to make the readers clear.

4. No significant differences were observed in accession 2 (2832.3) and 4 (5203.6) in Glucose, is the deviation too large?Check other data as well.

5. The quality of table 4 and figure 3 are low.

6. access 1 was lost in figure 4

Author Response

Please see the file sent as PDF. 

Reviewer 4 Report

Dear respected colleagues,

I carefully checked your manuscript and its data is suitable in this journal. Please amend the following queries before publishing this excellent manuscript. These comments help you to improve the quality of your manuscript and help academic readers to get significant clues easily.

1- Please add the figure of studied uvaia to the manuscript. If it is possible, please add the figure in the introduction section.

2- Please italicize all scientific names used in the manuscript script.

3- Please add another paragraph to the introduction section and try to elaborate on the botany, metabolic profile, and previous studies conducted on this tree that is in line with your research topic.

4- Please make a flowchart for your M&M section. That’s nice of you if the respected authors could cite all previous protocols used in this study.

5- Please check your reference list for problematic papers. Please make sure that validated references are cited in this case.

6- Please add geographical coordination of sampling sites.

7- Please turn numerical values detailed in Table 1 into a heatmap chart or similar graphs so that the differences can be understood easily.

8- Why secondary metabolites profile of studied accessions were not discussed in the paper? did the respected authors focus on primary metabolites?

9- In tables 3-4, the assigned words for each studied trait are a little confusing. Would you please determine which statistical methods have been used to elucidate the differences between each sample? The data represented in these tables can be generally expressed in comparative stacked bar plots. You can use libraries “tidyverse” and “viridis” in the R program to generate a comparative diagram for each studied fraction reported in these tables.

10- Please add a concise paragraph about the novelty of this work to your paper.

11- The respected authors mixed the discussion with the results. Please separate them and discuss the relevant literature based on your outcomes. In which traits your outcomes are varying the previous results reported in the literature? Please clearly discuss this case when you revise the paper.

12- which secondary metabolites were prominent metabolites in the studied samples? Would you please add the 2D structure of identified secondary metabolites to the supplementary files?

13- Please provide more explanation for the metabolic pathways illustrated in Figure 5.

14- Please improve the coherency of the manuscript text. It is a little bit confusing to understand the text and make a connection between different parts of your results. Please use active language to express your results.

15- How can these results be promoted practically to develop further studies in this line?

Author Response

Please see the file sent as PDF. 

Round 2

Reviewer 1 Report

The authors responded to all suggestions and the manuscript improved significantly.

Reviewer 4 Report

Dear authors, 

Thank you very much for your meticulous answers and perfect revision. The respected authors have carefully addressed all of my comments and I have no further comments on this paper. The current version is suitable for publication in this journal. Therefore, my recommendation is "acceptance" of this paper. 

Regards,